# The Optimal Daily Dispatch of Ice-Storage Air-Conditioning Systems

**Ching-Jui Tien and Ming-Tang Tsai ***

Department of Electrical Engineering, Cheng-Shiu University, Kaohsiung 833, Taiwan
* Correspondence: k0217@gcloud.csu.edu.tw; Tel.: +886-7-7310606

**Abstract:** In this paper, the Ant-based Radial Basis Function Network (ARBFN) is proposed to determine the optimal daily dispatch of ice-storage air-conditioning systems. ARBFN is a novel algorithm that is integrated into the Ant Colony Optimization and Radial Basis Function Network. ARBFN is used to construct the function of the cost and operation for each chiller and ice-storage tank and is used to simulate the polynomial function of the cooling load and the cost of power consumption. The best learning rate in the training process is adjusted in ARBFN to improve the accuracy of constructing models for chillers and ice-storage tanks. The electricity savings are thus 4.130% on a summer day and 7.381% on a non-summer day. The results have shown that ARBFN can more accurately calculate the actual power consumption and cooling capability of each chiller and ice-storage tank. Lastly, ACO is used to calculate the daily dispatch of the ice-storage air-conditioning system. The results demonstrated the optimization of energy savings and efficiency for the operation of the ice-storage air-conditioning system.

**Keywords:** ice-storage air-conditioning system; Radial Basis Function network; Ant Colony Optimization; energy saving

## 1. Introduction

The growth in electricity demand in recent years has become substantial and the summer peak load is rising year by year. Taiwan is an island and must generate 100% of its own electricity due to the fact that its power system is not connected to other power grids. Stable energy and a stable power supply are very important for Taiwan. The demand for electricity is continuously growing at an average of 3.2% per year and the capacity of air-conditioning systems occupies more than 30% of the overall energy consumption and up to 40% during peak periods [1]. Therefore, we have an urgent need to seek and develop alternative sources of energy and carry out comprehensive reviews of the efficiency of using these energy sources. Currently, public sites both in Taiwan and abroad have tended to grow larger in scale, and public buildings such as large hospitals, office buildings, and shopping malls must use air-conditioning systems. In many case studies, the cooling capacity is above 1000–10,000 RT, and the load is about 40–50% of the total power load. Ice-storage air-conditioning systems, which integrate chillers with an ice-storage tank, have been effectively applied to the management of the system cooling loads [2]. They shift the peak load of electricity consumption to off-peak hours in order to reduce the problem of overloads during peak hours, improve the efficiency of off-peak electricity use, and reduce peak electricity use. Therefore, power management in ice-storage air-conditioning systems is very important for energy saving [3].

In ice-storage air-conditioning systems, chillers account for the biggest share of the air-conditioning system power consumption, roughly 60% [4]. The electrical energy consumption of chiller plants markedly increases if the chillers are improperly managed. The optimization approach for multi-chiller systems has thus attracted significant attention. A benefit of the optimization approach is that it often leads to substantial savings in energy

consumption. Studies have been carried out on the effectiveness of optimal chiller loading (OCL) problems in the past [5–8]. Recently, some artificial intelligence techniques have been presented to solve the OCL problem and have shown their effectiveness [9,10]. Simulated annealing (SA) based on the kW/PLR curves includes non-convex functions to solve the OCL problem [11]. Additionally, a differential evolution algorithm (EP) was proposed to solve the OCL problem [12] and a two-stage differential evolution was used to solve the optimal chiller loading for energy saving [13]. Furthermore, a particle swarm optimization (PSO) algorithm was presented to solve the OCL problem and has shown its effectiveness [14,15]. The authors of [16] also proposed the Barnacles Mating Optimizer algorithm to optimize the chiller loading for energy conservation and reduction. Results indicated that an optimal dispatch of the chillers can be found so that the operating cost of the whole scheduling period can be minimized while satisfying numerous operating constraints. The problem becomes a difficult decision-making process because of the complex constraints so efficient tools are needed to determine the best dispatching system.

Ant colony optimization (ACO) is the activity characteristic of biotic populations [17,18]. The advantages of the ACO algorithm are that it learns and exchanges information to search for the shortest route between the colony and food sources. In the ACO method, individual solutions can converge to the optimal solution through a small number of evolution iterations. In this paper, the ant colony optimization algorithm is integrated with the radial basis function network to create the ant-based radial basis function network (ARBFN) for finding the optimal dispatch of ice-storage air-conditioning systems. The RBFN has many of the advantages of neural intelligence in searching, with an easy and fast convergence in computation, but it has poor stability in a higher-dimensional search. Because the ACO has the capability to optimize learning parameters, the best learning rate of the ARBFN is adjusted to improve the accuracy of constructing models. The ARBFN is then used to construct the function of the cost and operation of each chiller and ice-storage tank. The results showed that ARBFN can more accurately calculate the actual power consumption and cooling capability of each chiller and ice-storage tank. In addition, the ACO was used to calculate the daily dispatch of the ice-storage air-conditioning system and optimize the energy saving and efficiency of the operation of the ice-storage air-conditioning system. The simulation results from this study provided a novel tool to solve the problems of the economic dispatch of ice storage air-conditioning systems and can provide more efficient energy use to dispatch chillers for saving energy.

## 2. Problem Description

The ice-storage air-conditioning system operates ice-storage in the off-peak hours of electricity use. During peak electricity use, the ice is melted into water to release cold energy in order to meet the required cooling load. The ice-storage air-conditioning system fully utilizes the characteristics of chillers and ice-storage tanks in order to provide better operations and electricity planning. In a conventional ice-storage air-conditioning system, the energy-saving planning of the chiller groups and the ice-storage tank is conducted mainly by letting the chiller groups be responsible for supplying the whole day's cooling load for the target space. The mathematical formulas of the chillers and ice-storage tank are introduced as follows:

### 2.1. The Cooling Load of the Chiller

The cooling load of the chillers in the air-conditioning system usually considers the measured return and supply water temperature of the chiller units and the cooling water flow to calculate the cooling load of the chiller using (1) to (3):

$$\Delta T_{chw,i} = T_{chwrt,i} - T_{chwst,i} \tag{1}$$

$$\Delta T_{chw,i\mathrm{min}} \leq \Delta T_{chw,i} \leq \Delta T_{chw,i\mathrm{max}} \tag{2}$$

$$Q_{chiller,i} = \left(LPM_{chiller,i} \times (\Delta T_{chw,i}) \times \rho_w \times C_{pw}\right) \times 60 \tag{3}$$

### 2.2. Ice-Storage Cooling System

The operation modes of the ice-storage cooling system are divided into two kinds: "charge process" and "discharge process". The ice-storage operation during off-peak periods freezes water into ice in order to store the cooling capability, and the cooling load capability of ice melting is calculated with water temperature differences and the water flow valve. Thus, the cooling load of the ice-storage cooling system is calculated by (4) to (7):

$$\Delta T_{isw} = T_{iswr} - T_{isws} \tag{4}$$

$$\Delta T_{isw,\min} \leq \Delta T_{isw} \leq \Delta T_{isw,\max} \tag{5}$$

$$Q_{ice} = \left(\Delta T_{isw} \times LPM_{ice} \times C_{pw} \times \rho_w\right) \times 60 \tag{6}$$

$$LPM_{ice\min} \leq LPM \leq LPM_{ice\max} \tag{7}$$

## 3. Solution Algorithms

### 3.1. The Models of Chillers and Ice-Storage Tanks

The chiller function can be expressed (8) [5]:

$$P_{chiller,i} = a_i + b_i Q_{chiller,i} + c_i Q_{chiller,i}^2 + d_i Q_{chiller,i}^3 \tag{8}$$

where $P_{chiller,i}$ is the power consumption of the $i$-th chiller and $a_i$, $b_i$, $c_i$, $d_i$ are the regression coefficients of the $i$-th chiller. The model constructed with traditional methods tends toward linear functions and mainly depends on the $T_{chwrt,i}$ return and $T_{chwst,i}$ supply water temperature of the ice-storage air-conditioning system. The charge and discharge process of the ice-storage tank are defined as (9) and (10), respectively:

$$ICE_{cp} = a_{ice,cp} + b_{ice,cp} P_{ice} + c_{ice,cp} P_{ice}^2 + d_{ice,cp} P_{ice}^3 \tag{9}$$

$$ICE_{dp} = a_{ice,dp} + b_{ice,dp} Q_{ice} + c_{ice,dp} Q_{ice}^2 + d_{ice,dp} Q_{ice}^3 \tag{10}$$

where $a_{ice,cp}$, $b_{ice,cp}$, $c_{ice,cp}$, and $d_{ice,cp}$ are regression coefficients of the function of the $ICE_{cp}$ ice-storage capacity and $P_{ice}$ is the power consumption cost. The cooling load capacity of ice melting is calculated based on the amount of melting ice.

The actual controllable parameters of the chillers and ice-storage tank are used in order to demonstrate the association between the power capacity of the chillers and ice-storage tank and the cooling capacity of the system. The main purpose of this paper is to find the optimal dispatch of the ice-storage air-conditioning system that satisfies the demanded cooling load of the system. The objective function is defined as (11):

$$Objective\ function = \sum_{t=1}^{h} \left( \sum_{i=1}^{l} P_{chiller,i}^t U_i^t \times Price_{chiller}^t + P_{ice}^t \times Price_{ice}^t \right) \tag{11}$$

The cooling conditions of the target space as shown in (12) must also be satisfied.

$$\left( \sum_{i=1}^{l} Q_{chiller,i}^t + Q_{ice}^t \right) \geq CL^t \tag{12}$$

where $Q^t_{chiller,i}$ is the cooling load of *i*-th chiller during the hour $t$ (kJ/h), $Q^t_{ice}$ is the cooling load of *i*-th ice-storage during the hour $t$ (kJ/h), and $CL^t$ is the system total cooling load during the hour $t$ (kJ/h).

### 3.2. Ant-Based Radial Basis Function Network (ARBFN)

The ARBFN consists of the input, hidden, and output layers. The signal propagation and the basic function of each layer are described in [19]. The ARBFN structure is shown in Figure 1 and the "ACO" process is performed in the training stage. The learning rates $\mu_w, \mu_c, \mu_\sigma$ are adjusted by ACO as stated below. The ACO technique finds the optimal solution using a population of ants. The population size is set to $P = 20$ and the dimension is set to $d = 3$. The generation ants are $\mu^d_i = [\mu_w, \mu_c, \mu_\sigma]$, where $\mu_w$, $\mu_c$ and $\mu_\sigma$ are the ARBFN learning rates. To evaluate the accuracy of the ARBFN in testing, the mean absolute percentage error (MAPE), as calculated with (13), is used in this paper.

$$MAPE = \frac{1}{2U} \sum_{u=1}^{U} \left( \frac{\left| P^{ture}_u - P^{cal}_u \right|}{P^{ture}_u} + \frac{\left| Q^{ture}_u - Q^{cal}_u \right|}{Q^{ture}_u} \right) \times 100\% \tag{13}$$

where $P^{ture}_u$ and $Q^{ture}_u$ are the actual data of the chiller and ice-storage tank, $P^{cal}_u$ and $Q^{cal}_u$ are the ARBFN regression data, and $U$ is the number of testing data.

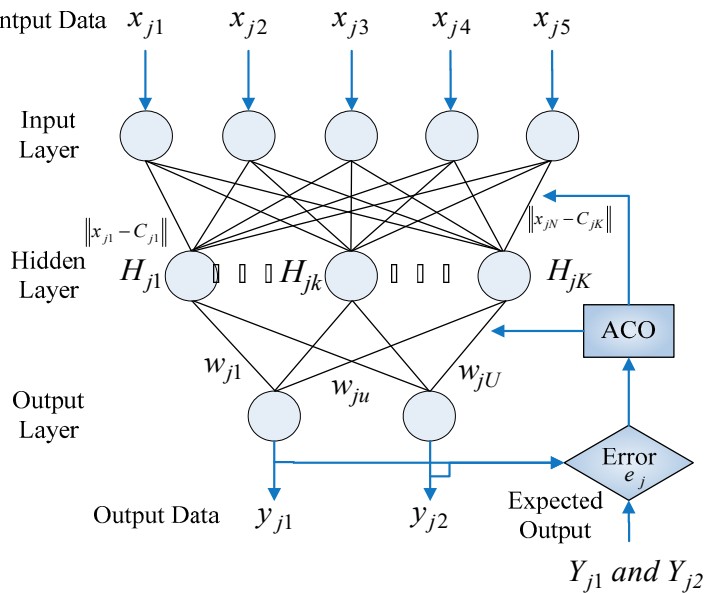

**Figure 1.** The ARBFN structure *j*-th model.

This study mainly adopts the ARBFN to find various operating parameters of chillers and ice-storage tanks. The cost-performance curve for each unit was simulated in order to solve the energy-saving planning for the ice-storage air-conditioning system. The operation of the ice-storage air-conditioning system and all the chillers and ice-storage tanks is an optimization problem. Therefore the scheduling strategy adopted for energy-saving in the ice-storage air-conditioning system is to determine the operating parameter settings of each chiller and ice-storage tank in each time period. The ant colony optimization algorithm has been verified on complex problems. It not only possesses excellent efficiency, but it can also convert the presentation of the problem-solving process into paths. Through probability calculation and updating pheromones, the best path is selected. Therefore, when encountering several local optima, it can find better objective functions than conventional search algorithms.

The ARBFN is used to construct the function of cost and operation for each chiller and ice-storage tank. The operating data of the charge process and discharge process of

each chiller and ice-storage tank are collected and divided into input variables and output variables. Then, the cooling load and power consumption for each chiller and ice-storage tank can be constructed. ARBFN is used to construct the cost function for each chiller and ice-storage tank. Therefore, when the input variables of the chiller are determined, the corresponding output variables can be obtained through the ARBFN.

The chillers' net is expressed as

Input = [Temp, $T_{chwrt}$, PLR, $\Delta T_{chw}$, $\Delta T_{cw}$]

Output = [$P_{chiller}$, $Q_{chiller}$]

The charge process of the ice-storage net is expressed as

Input = [Temp, $\Delta T_{isw}$, $\Delta T_{bcw}$, $\Delta T_{bccw}$, Ice Capacity (%)]

Output = [$P_{ice}$, Ice Charge (RT)]

The discharge process of the ice-storage net is expressed as

Input = [Temp, $T_{iswr}$, $\Delta T_{isw}$, LPM, Ice Capacity (%)]

Output = [$Q_{ice}$, Ice Discharge (RT)]

The operation of the ice-storage net is expressed by (14) and (15)

$$\Delta T_{bcw} = T_{bcws} - T_{bcwr} \tag{14}$$

$$\Delta T_{bccw} = T_{bccws} - T_{bccwr} \tag{15}$$

The steps of ARBFN are as follows:

(1) Use the ACO algorithm to compute the 24-h switching state of each chiller.
(2) The operating parameters of each chiller are used by the NO.1 to NO.6 chiller net to calculate their individual power consumption and cooling load capacity. For the ice-storage tank, the ice-storage operation consists of a total of nine hours from 22:00 to 06:00 the next day. The operating parameters are used by the charging process net to calculate the power consumption and ice-storage volume. From 07:00 to 21:00, the ice-melting operation occurs for a total of 15 h. The operating parameters are then used by the discharge process net to calculate the cooling load capacity and the amount of melting ice.
(3) The sum of the cooling load capacity of all chillers and ice-storage tanks should also meet the required cooling load of the system in each time period. The total power consumption multiplied by the electricity price per time is the sum of the total cost.

## 4. Simulation Results

In this paper, six chillers (NO.1 and NO.2 are sets of 550RT and NO.3 to NO.6 are sets of 1000RT) and ice-storage tank sets of 8000RT are used as the simulation case. The power price for the chillers and ice-storage tanks is calculated based on the time-of-use rate in [20]. The required hourly data of the ice storage system were collected from 22:00 of the previous day until 21:00 of the following day on two days (a summer day and non-summer day).

### 4.1. Least-Squares Regression (LSR) Model Verification

The collected data are used to plot the distribution of $P_{chiller}$ power consumption and $Q_{chiller}$ cooling capacity. Figure 2 shows that although the power consumption and cooling capacity data of the NO.3 chiller are chaotically scattered, they generally exhibit a proportional rising. Equation (8) is used in this paper to construct the third-order polynomial function of power consumption using LSR. Therefore, Figure 2 shows that the relationship between $P_{chiller}$ and $Q_{chiller}$ is almost linear. The constructed corresponding output function is shown in Figure 3 and Table 1 shows the coefficients of the polynomial functions for chillers and ice-storage tanks constructed by LSR.

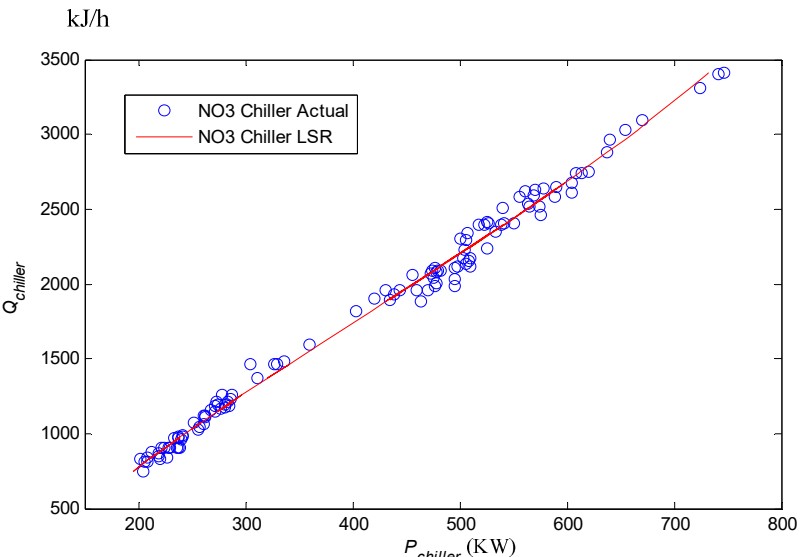

**Figure 2.** P-Q curve of chiller NO.3.

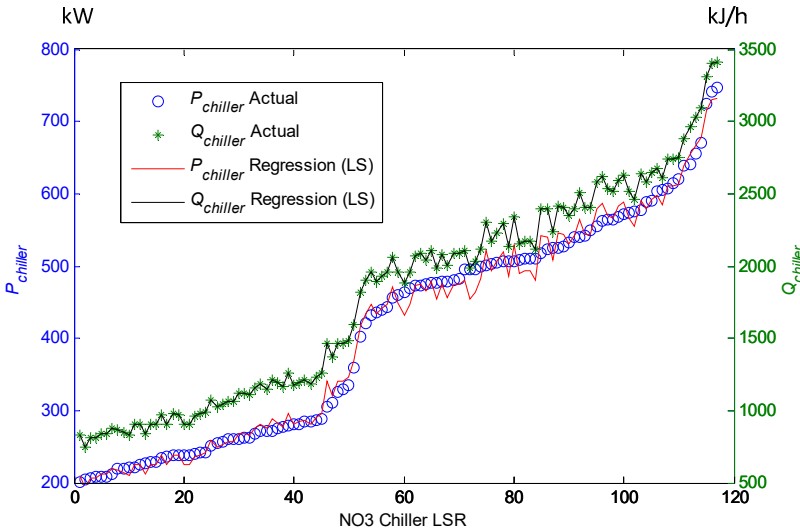

**Figure 3.** Chiller NO.3 in the LSR model.

**Table 1.** Chillers NO.1-NO.6 and the ice storage LSR.

| Unit | a | b | c | d |
|:---:|:---:|:---:|:---:|:---:|
| $P_{chiller,1}$ | 65.7772 | 0.196085 | $1.3707 \times 10^{-8}$ | $1.249 \times 10^{-9}$ |
| $P_{chiller,2}$ | 128.7969 | 0.044904 | 0.000113908 | $-2.628 \times 10^{-8}$ |
| $P_{chiller,3}$ | 68.2033 | 0.141784 | $4.13921 \times 10^{-5}$ | $-7.599 \times 10^{-9}$ |
| $P_{chiller,4}$ | 107.7250 | 0.118114 | $1.87115 \times 10^{-5}$ | $-1.467 \times 10^{-9}$ |
| $P_{chiller,5}$ | 623.2087 | $-0.455524$ | 0.000228205 | $-2.660 \times 10^{-8}$ |
| $P_{chiller,6}$ | 101.5365 | 0.085082 | $6.87455 \times 10^{-5}$ | $-1.141 \times 10^{-8}$ |
| $ICE_{cp}$ | 2204.5246 | $-24.353361$ | 0.092520429 | $-0.0001022$ |
| $ICE_{dp}$ | $-21.7173$ | 0.220563 | $5.53312 \times 10^{-5}$ | $-1.591 \times 10^{-8}$ |

*4.2. ARBFN Model Verification*

Table 2 shows the learning performance of the ARBFN, RBFN, and BPN for comparison of the NO.3 chiller net. With the regulation of learning rates $\mu_w$, $\mu_c$, and $\mu_\sigma$ by ACO, the ARBFN has a better accuracy than the other methods in Table 2 and is shown in Figure 4.

**Table 2.** Performance comparison with various control methods in the NO.3 chiller net.

| Method | Number of Training Data | Number of Test Data | MAPE (%) | Number of Training Data | Number of Test Data | MAPE (%) |
|--------|-------------------------|---------------------|----------|-------------------------|---------------------|----------|
| ARBFN | 117 | 11 | 1.062 | 106 | 22 | 2.048 |
| RBFN | 117 | 11 | 2.431 | 106 | 22 | 4.779 |
| BPN | 117 | 11 | 4.679 | 106 | 22 | 8.547 |

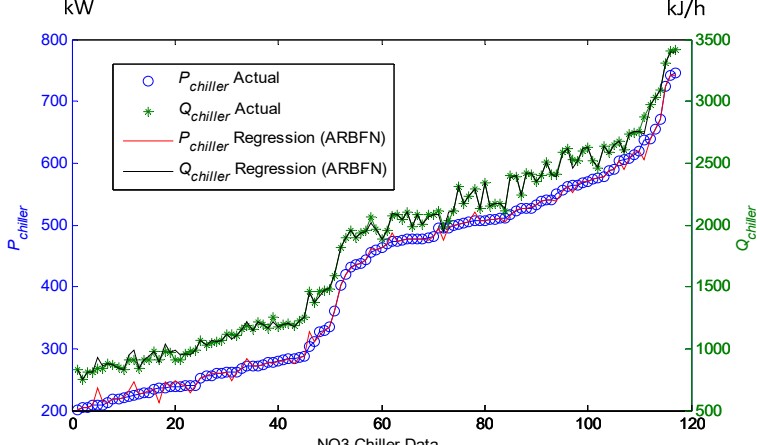

**Figure 4.** Chiller NO.3 on ARBFN.

The ARBFN is used to construct the function model of the hourly ice storage of the charge process net for the ice-storage tank as shown in Figure 5. The ARBFN is used to construct the function model of the hourly ice melting of the discharge process net for the ice-storage tank, and the association between the cooling capability and ice discharge is shown in Figure 6. If the melting ice volume of the ice-storage tank $Q_{ice}$ decreases, the cooling load will also decrease.

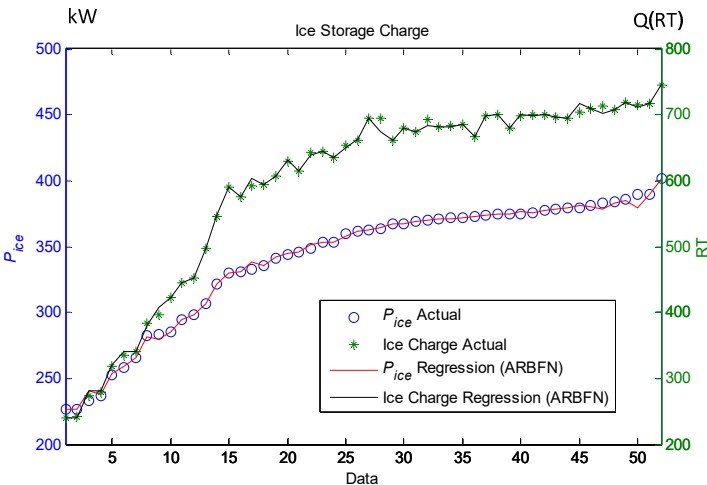

**Figure 5.** Charge process net.

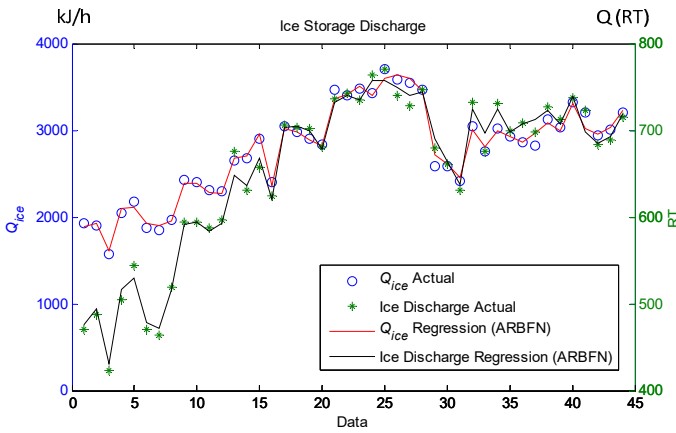

**Figure 6.** Discharge process net.

### 4.3. The Dispatch of ICE-Storage Air-Conditioning System

　　The condition parameters on a summer day and non-summer day, as given in Figures 7 and 8, are simulated. The hourly required cooling capacity, outside air temperature, and $T_{chwrt}$ of the ice-storage system were collected from 22:00 of the previous day, until 21:00 of the following day on both a summer and non-summer day.

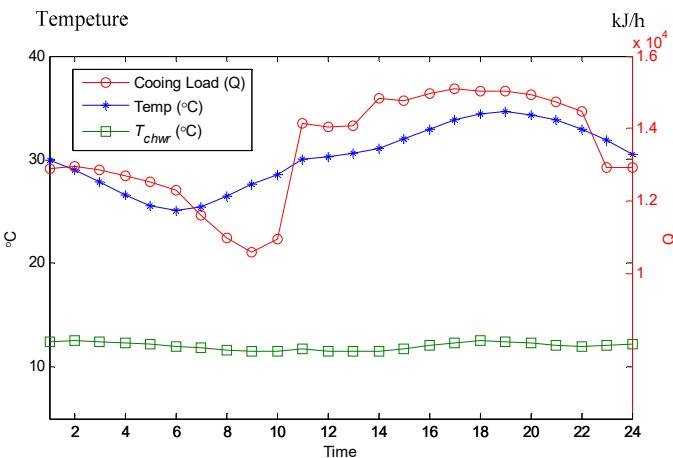

**Figure 7.** System condition parameters on a summer day.

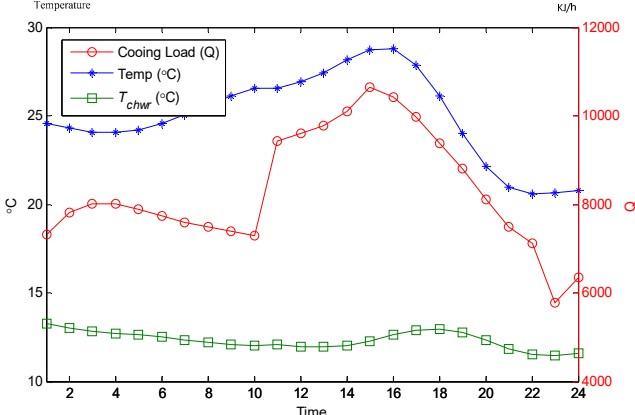

**Figure 8.** System condition parameters on a non-summer day.

　　In this paper, the accuracy of the LSR and ARBFN models for the chiller and ice-storage tank is analyzed with the system cooling load on a summer day and a non-summer day. The parameters of each unit were used as input data for LSR and ARBFN to compute the

cooling capacity and power consumption in order to calculate the cost of a single day's power consumption. Finally, as given in Tables 3 and 4, the actual power consumption measured by the ice-storage air-conditioning system was compared for verification. In the tables, the "actual" data are the actual measurements of the ice-storage air-conditioning system. The power consumption of a single day was also listed for comparative analysis.

**Table 3.** Comparative analysis of power consumption on a summer day.

| Hour | Actual | ARBFN | | LSR | |
|---|---|---|---|---|---|
| | Power (kW) | Power (kW) | [1] Error (%) | Power (kW) | [2] Error (%) |
| 22 | 3023.882 | 3073.545 | 1.64 | 3336.999 | 10.35 |
| 23 | 3224.135 | 3114.188 | 3.41 | 3380.314 | 4.84 |
| 24 | 3213.706 | 3116.380 | 3.03 | 3339.555 | 3.92 |
| 1 | 3145.202 | 3083.166 | 1.97 | 3324.830 | 5.71 |
| 2 | 3179.706 | 3041.616 | 4.34 | 3326.815 | 4.63 |
| 3 | 3131.653 | 3031.975 | 3.18 | 3298.529 | 5.33 |
| 4 | 2745.719 | 2882.666 | 4.99 | 3060.697 | 11.47 |
| 5 | 2828.079 | 2761.755 | 2.35 | 2940.212 | 3.97 |
| 6 | 2641.276 | 2673.040 | 1.20 | 2791.609 | 5.69 |
| 7 | 2024.825 | 1928.986 | 4.73 | 2168.484 | 7.09 |
| 8 | 2583.937 | 2595.920 | 0.46 | 2892.467 | 11.94 |
| 9 | 2608.795 | 2544.626 | 2.46 | 2824.032 | 8.25 |
| 10 | 2447.217 | 2478.641 | 1.28 | 2610.111 | 6.66 |
| 11 | 2446.248 | 2516.830 | 2.89 | 2660.329 | 8.75 |
| 12 | 2406.625 | 2530.431 | 5.14 | 2673.871 | 11.10 |
| 13 | 2828.901 | 2708.706 | 4.25 | 3027.895 | 7.03 |
| 14 | 2728.399 | 2619.541 | 3.99 | 2778.751 | 1.85 |
| 15 | 2671.204 | 2606.755 | 2.41 | 2846.374 | 6.56 |
| 16 | 3409.064 | 3248.773 | 4.70 | 3490.559 | 2.39 |
| 17 | 3130.662 | 3160.913 | 0.97 | 3468.779 | 10.80 |
| 18 | 3122.562 | 3205.378 | 2.65 | 3424.233 | 9.66 |
| 19 | 3208.062 | 3069.716 | 4.31 | 3362.217 | 4.81 |
| 20 | 2816.290 | 2741.367 | 2.66 | 3012.843 | 6.98 |
| 21 | 2915.637 | 2827.455 | 3.02 | 3014.334 | 3.39 |
| Total (kW) | 68,481.79 | 67,562.37 | 1.34 | 73,054.84 | 6.68 |
| Cost NT$ | 194,726 | 192,310 | 1.24 | 181,517 | 6.78 |

[1]$Error = \frac{|Actual(Power) - ARBFN(Power)|}{Actual(Power)} \times 100\%$; [2]$Error = \frac{|Actual(Power) - LSR(Power)|}{LSR(Power)} \times 100\%$.

In this paper, the errors between the actual power consumption of the chillers and ice-storage tank and the estimations of the ARBFN and LSR were compared and shown in Tables 3 and 4. The maximum error of ARBFN in a single hour on a summer day was 5.14% and 5.13% in a single hour on a non-summer day, while the maximum error of LSR in a single hour on a summer day was as high as 11.94% and in a single hour on a non-summer day was as high as 28.64%. The average hourly error of the ARBFN on a summer day and a non-summer day was 2.815%, and the difference in power consumption was 50.10 kW. The average hourly error of LSR was 7.805%, and the difference in power consumption was 144.81 kW. Therefore, the accuracy of the ARBFN models can be verified. If LSR is chosen

for the scheduling assessment of the ice-storage air-conditioning system, larger errors in costs tend to occur.

**Table 4.** Comparative analysis of power consumption on a non-summer day.

| Hour | Actual | ARBFN | | LSR | |
|---|---|---|---|---|---|
| | Power (kW) | Power (kW) | [1] Error (%) | Power (kW) | [2] Error (%) |
| 22 | 2071.718 | 2019.278 | 2.53 | 2004.281 | 3.26 |
| 23 | 2183.114 | 2113.548 | 3.19 | 2111.880 | 3.26 |
| 24 | 2151.736 | 2172.034 | 0.94 | 2163.087 | 0.53 |
| 1 | 2048.908 | 2134.770 | 4.19 | 2177.811 | 6.29 |
| 2 | 2022.306 | 2085.315 | 3.12 | 2116.379 | 4.65 |
| 3 | 2087.035 | 2093.330 | 0.30 | 2113.926 | 1.29 |
| 4 | 1984.494 | 2086.231 | 5.13 | 2070.827 | 4.35 |
| 5 | 2081.175 | 2030.614 | 2.43 | 1968.841 | 5.40 |
| 6 | 1953.010 | 1955.428 | 0.12 | 1936.210 | 0.86 |
| 7 | 1234.242 | 1184.907 | 4.00 | 914.504 | 25.91 |
| 8 | 1535.591 | 1495.657 | 2.60 | 1346.210 | 12.33 |
| 9 | 1540.546 | 1591.767 | 3.32 | 1546.554 | 0.39 |
| 10 | 1566.713 | 1594.757 | 1.79 | 1407.467 | 10.16 |
| 11 | 1688.817 | 1669.943 | 1.12 | 1672.489 | 0.97 |
| 12 | 1693.599 | 1711.994 | 1.09 | 1594.723 | 5.84 |
| 13 | 1803.121 | 1843.776 | 2.25 | 1885.726 | 4.58 |
| 14 | 1585.353 | 1653.696 | 4.31 | 1413.713 | 10.83 |
| 15 | 1472.108 | 1537.330 | 4.43 | 1328.556 | 9.75 |
| 16 | 1596.275 | 1671.406 | 4.71 | 1441.107 | 9.72 |
| 17 | 1435.071 | 1447.220 | 0.85 | 1050.250 | 26.82 |
| 18 | 1423.891 | 1417.715 | 0.43 | 1078.923 | 24.23 |
| 19 | 1395.253 | 1434.252 | 2.80 | 995.708 | 28.64 |
| 20 | 1409.325 | 1356.831 | 3.72 | 1313.006 | 6.83 |
| 21 | 1493.497 | 1438.046 | 3.71 | 1426.941 | 4.46 |
| Total (kW) | 41,456.900 | 41,739.850 | 0.68 | 39,079.120 | 5.74 |
| Cost (NT$) | 91,457 | 92,249 | 0.87 | 98,605 | 7.25 |

[1]$Error = \frac{|Actual(Power) - ARBFN(Power)|}{Actual(Power)} \times 100\%$; [2]$Error = \frac{|Actual(Power) - LSR(Power)|}{LSR(Power)} \times 100\%$.

Table 5 shows the operating status of chillers during 24 h periods on a summer day and a non-summer day. From Table 5, during off-peak hours when the cooling load is smaller, the ice maker stores the required cooling energy in the storage tank. During peak hours, the storage tank provides the required cooling load. Figure 9 shows the operating cost of chillers during 24 h periods on a summer day and a non-summer day. It can be seen in Figure 9 that the TOU rate will influence the overall economy of the ice storage air-conditioning system.

**Table 5.** The operating status of chillers on a summer day and non-summer day.

| Hour | Summer Day | | | | | | | Non-Summer Day | | | | | |
|---|---|---|---|---|---|---|---|---|---|---|---|---|---|
| | No.1 | No.1 | No.2 | No.3 | No.4 | No.5 | No.6 | No.1 | No.2 | No.3 | No.4 | No.5 | No.6 |
| 1 | 0 | 1 | 1 | 1 | 1 | 1 | 0 | 0 | 1 | 0 | 1 | 1 | 1 |
| 2 | 1 | 1 | 1 | 0 | 1 | 1 | 1 | 1 | 0 | 0 | 1 | 0 | 1 |
| 3 | 1 | 1 | 1 | 1 | 1 | 0 | 1 | 1 | 0 | 1 | 1 | 0 | 1 |
| 4 | 0 | 1 | 1 | 1 | 1 | 0 | 0 | 1 | 1 | 0 | 1 | 1 | 1 |
| 5 | 0 | 1 | 0 | 1 | 1 | 1 | 0 | 1 | 1 | 0 | 1 | 0 | 1 |
| 6 | 0 | 1 | 1 | 0 | 1 | 1 | 0 | 1 | 0 | 1 | 1 | 1 | 0 |
| 7 | 0 | 0 | 1 | 1 | 1 | 0 | 0 | 1 | 0 | 0 | 0 | 0 | 1 |
| 8 | 0 | 0 | 1 | 1 | 1 | 1 | 0 | 0 | 0 | 1 | 0 | 1 | 1 |
| 9 | 0 | 0 | 0 | 1 | 1 | 1 | 0 | 0 | 1 | 1 | 1 | 0 | 1 |
| 10 | 0 | 0 | 1 | 1 | 1 | 1 | 0 | 0 | 0 | 1 | 1 | 1 | 0 |
| 11 | 1 | 1 | 1 | 0 | 1 | 1 | 1 | 0 | 0 | 1 | 1 | 1 | 0 |
| 12 | 1 | 0 | 1 | 1 | 1 | 0 | 1 | 0 | 1 | 1 | 1 | 1 | 0 |
| 13 | 0 | 1 | 1 | 1 | 1 | 1 | 0 | 0 | 0 | 0 | 1 | 1 | 1 |
| 14 | 1 | 0 | 1 | 1 | 1 | 0 | 1 | 0 | 1 | 1 | 0 | 1 | 0 |
| 15 | 1 | 0 | 1 | 1 | 1 | 1 | 1 | 0 | 0 | 1 | 1 | 1 | 0 |
| 16 | 1 | 0 | 1 | 1 | 1 | 1 | 1 | 0 | 0 | 1 | 1 | 1 | 0 |
| 17 | 1 | 0 | 1 | 1 | 1 | 1 | 1 | 1 | 0 | 1 | 0 | 0 | 1 |
| 18 | 0 | 1 | 1 | 1 | 1 | 1 | 0 | 0 | 0 | 1 | 1 | 0 | 0 |
| 19 | 0 | 1 | 1 | 1 | 1 | 1 | 0 | 0 | 0 | 1 | 1 | 1 | 0 |
| 20 | 1 | 1 | 1 | 1 | 0 | 1 | 1 | 0 | 0 | 1 | 0 | 0 | 1 |
| 21 | 0 | 0 | 1 | 1 | 1 | 1 | 0 | 0 | 0 | 0 | 1 | 1 | 0 |
| 22 | 1 | 1 | 1 | 0 | 1 | 1 | 1 | 0 | 1 | 0 | 1 | 1 | 1 |
| 23 | 1 | 1 | 0 | 1 | 1 | 1 | 1 | 1 | 1 | 1 | 1 | 0 | 1 |
| 24 | 0 | 0 | 1 | 1 | 1 | 1 | 0 | 0 | 1 | 1 | 1 | 1 | 1 |

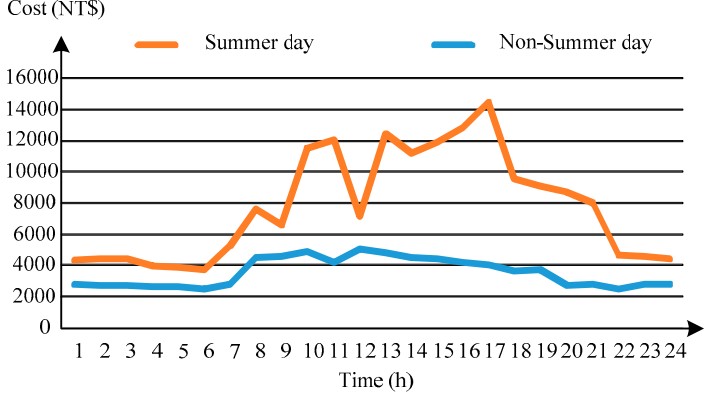

**Figure 9.** The operating cost of chillers on a summer day and non-summer day.

The convergent characteristics of operating costs on a summer day (17 July) and a non-summer day (21 October) are as given in Figure 10. Using ACO to optimize the energy planning for the ice-storage air-conditioning system can effectively reduce costs. The electricity savings are thus 4.130% on a summer day and 7.381% on a non-summer day. This system can also yield a better plan for ice storage and melting procedures.

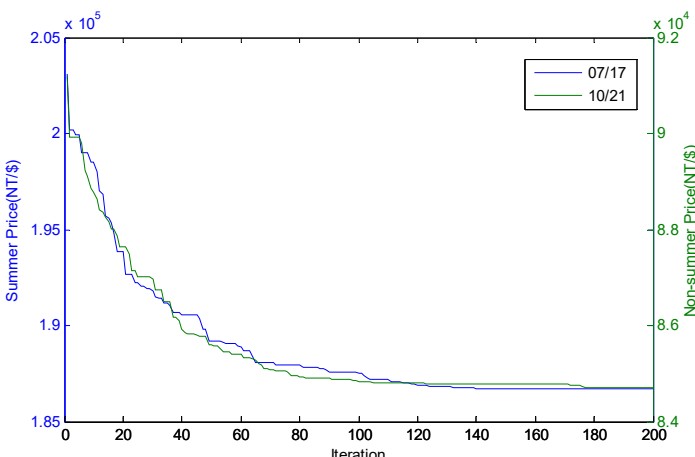

**Figure 10.** Convergent characteristics of operating cost on a summer day and non-summer day.

Table 6 shows the comparisons of the various algorithms. An IBM PC with a P-IV2.0 GHz CPU and 512 MB SDRAM was used for this test. The improvement of the ARBFN over other algorithms is clear. Although the execution time of the ARBFN was longer, it did indicate the capacity of ARBFN to discover a better optimum.

**Table 6.** Comparison of the various algorithms.

| | Summer Day | | Non-Summer Day | |
|---|---|---|---|---|
| **Algorithms** | **Total Cost (NT$)** | **Execution Time (s)** | **Total Cost (NT$)** | **Execution Time (s)** |
| ARBFN | 186683.17 | 5.67 | 84706.24 | 5.67 |
| GA-RBFN | 187309.84 | 4.81 | 85371.62 | 4.81 |
| EP-RBFN | 188131.15 | 3.54 | 85984.38 | 3.54 |

GA-RBFN: GA with RBFN EP-RBFN: EP with RBFN.

## 5. Conclusions

In this paper, the ARBFN is combined with the ACO algorithm to determine the optimal planning of the ice-storage air-conditioning system. The advantages of the ARBFN in regression analysis include simulating the corresponding power consumption and cooling capacity of each chiller and ice-storage tank, thus providing their operating parameters. Using the ACO algorithm for the parameter learning of the radial basis function network keeps the training speed within an allowable range, and it is more accurate than traditional polynomial regression methods in modeling the function of power consumption and cooling capacity. The ACO algorithm is also integrated to derive energy-saving planning for the ice-storage air-conditioning system. In this paper, actual cases were used to verify the effectiveness of the proposed method. For future development, it can be extended to more restricted types, such as an efficiency analysis for the operation and continuous operation of ice-storage systems with variable frequency as well as adjustment plans for ice-storage and melting in ice-storage systems. It is expected that these results will be more in line with the energy-saving planning of the ice-storage air-conditioning system. This method could also improve the operating efficiency of air-conditioning equipment.

**Author Contributions:** C.-J.T. is the first author. He provided the project idea, related experiences, and system model, and revised the English. M.-T.T. assisted with the project and prepared the manuscript as the corresponding author. All authors discussed the simulation results and approved the publication. All authors have read and agreed to the published version of the manuscript.

**Funding:** This research was supported by the National Science and Technology Council, Taiwan. (Grant Nos. MOST 111-2221-E-230-002).

**Institutional Review Board Statement:** Not applicable.

**Informed Consent Statement:** Not applicable.

**Data Availability Statement:** Not applicable.

**Conflicts of Interest:** The authors declare no conflict of interest.

## Nomenclature

| | |
|---|---|
| $C_{pw}$ | The specific heat of chilled water (4.186 kJ/kg) |
| $LPM_{chiller,i}$ | Liter per minute of chilled water, 1 RT = 10 LPM |
| $LPM_{ice}$ | Liter per minute of ice-storage water control valve |
| $LPM_{ice\min}$ | The liters per minute lower bound of ice-storage water |
| $LPM_{ice\max}$ | The liters per minute upper bound of ice-storage water |
| $P_{chiller,i}$ | The power consumption of the chiller (kW) |
| $P_{chiller,i}^{t}$ | The power consumption of the $i$-th chiller during hour $t$ |
| $Price_{chiller}^{t}$ | The power price of a chiller during hour $t$ |
| $P_{ice}^{t}$ | The power consumption of the ice storage during hour $t$ |
| $Price_{ice}^{t}$ | The power price of ice storage during hour $t$ |
| $Q_{ice}$ | Cooling load of the ice storage (kJ/h) |
| $T_{iswr}$ | The return temperature of the ice-storage water (°C) |
| $T_{isws}$ | The supply temperature of the ice-storage water (°C) |
| $T_{chwst,i}$ | The supply temperature of chilled water (°C) |
| $T_{bccwr}$ | The return temperature of brine chiller cooling water (°C) |
| $T_{bccws}$ | The supply temperature of brine chiller cooling water (°C) |
| $T_{chwrt,i}$ | The return temperature of chilled water (°C) |
| $T_{bcwr}$ | The return temperature of brine chiller water (°C) |
| $T_{bcws}$ | The supply temperature of brine chiller water (°C) |
| $U_{i}^{t}$ | The $i$-th chiller on/off during the hour $t$ |
| $\Delta T_{chw,i}$ | The temperature difference of chilled water (K) |
| $\Delta T_{chw,i\min}$ | The temperature differences of the lower bound of chilled water (°C) |
| $\Delta T_{chw,i\max}$ | The temperature differences of the upper bound of chilled water (°C) |
| $\Delta T_{isw}$ | The temperature difference of ice-storage water (°C) |
| $\Delta T_{isw,\min}$ | The temperature differences of the lower bound of ice-storage water (°C) |
| $\Delta T_{isw,\max}$ | The temperature differences of the upper bound of ice-storage water (°C) |
| $\Delta T_{bcw}$ | The temperature difference of brine chiller water (°C) |
| $\Delta T_{bccw}$ | The temperature difference of brine chiller cooling water (°C) |
| $\rho_{w}$ | The density of chilled water (1 kg/L) |

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
