# Peer review of "The Optimal Daily Dispatch of Ice-Storage Air-Conditioning Systems"

_inventions, doi:10.3390/inventions8020062_

Round 1

Reviewer 1 Report

Comments

In this paper, Ant-based Radial Basis Function Network (ARBFN) is proposed to find the optimal daily dispatch of ice-storage air-conditioning systems. The topic of this research is very interesting I would suggest the publication of this article if the following minor issues were addressed. 

-       The introduction is very brief, more recent works should be added related to energy storage. Authors may include the closely related works such as:

o   Ben Khedher, N. 2018. “Numerical Study of the Thermal Behavior of a Composite Phase Change Material (PCM) Room”. Engineering, Technology & Applied Science Research 8 (2). Greece:2663-67. https://doi.org/10.48084/etasr.1824.

o   Khedher, N. Ben, & Nasrallah, S. Ben. (2011). THREE-DIMENSIONAL SIMULATION OF A POROUS THERMAL ENERGY STORAGE SYSTEM USING SOLID-LIQUID PHASE CHANGE MATERIAL. Journal of Porous Media, 14(9), 777–790. https://doi.org/10.1615/JPORMEDIA.V14.I9.30

-       It is convenient to include a Nomenclature. Since many symbols are used in different parts of the paper.

-       In eq 6, the density of water is missing because mass flow rate should be considered to obtain the unit KJ/h of Qice

-       Figure 3, Please add the unit to the axis labels.

-       In Table 5 The operating status of chillers on the summer day and non-summer day, first revise status in the table caption. Secondly the on off status is automatically chosen or this a random choice for this study. More details should be given on the choice of the On/ OFF status of the chillers.

-       Figure 9, please replace summary day by summer day in legend.

-       More details on the used software to implement the ARBFN combined with the ACO algorithm.

-       Section 4, Authors mentioned that “The collected data are used to plot the distribution of chiller P power consumption and Q cooling capacity” more details are needed on the period of these data and how it is collected.

-       Line 257, it is mentioned that “ARBFN has a better accuracy than other methods in Table 3 and shown in Figure 4.” Please check the table number, it is table 2 and not 3.

-       Line 263, “the ice-storage volume is as shown in Figure 5.” However, the figure does not depict directly the volume of storage tank.

Author Response

Dear Reviewer:

Thank you for providing us the review’s comments. We have taken care of these precious comments and revised manuscript with the changes clearly identified by a highlighter pen. I sincerely hope that we have clarified all your questions. Your assistance is very much appreciated. If you have further questions, please feel free to contact me.

Your assistance is highly appreciated.

Sincerely yours.

Dr. Ming-Tang Tsai

Department of E.E.,

Cheng-Shiu University

Email:k0217@gcloud.csu.edu.tw

Reviewer 2 Report

The work presented for review, entitled The Optimal Daily Dispatch of Ice-Storage Air-Conditioning Systems, deals with the problem of controlling the operation of air-conditioning systems with ice storage. The work uses the ARBFN network for this purpose. The issue addressed from a practical point of view is important. Optimization of the control process brings both financial and social benefits.
Despite the merits of the work, I have several criticisms:
1.The abstract does not contain specific research results. I think it is too general and does not encourage to read the whole work.
 2. The review of the literature does not clearly indicate the need for further research in this area. Please indicate the disadvantages of the methods used in the process of controlling the operation of systems with air conditioning.
3. Indicate what is new in this work.
4. line 26 -28 - Are these the results of your own research? If not provide the source of the information.
5. line 34 - Perhaps it would be better to state the power in W and not in RT?
6. line 78 - Has the quality of the algorithm been analyzed for different dynamics of changes in heating and cooling demand and availability of energy from the grid? Does the model work only for fixed time zones? Can the model account for changes in energy prices dynamically? (We have an energy market and the price at any given time depends on the availability and demand of energy).
7. At what intervals were the supply and return water temperatures and flow rates measured?
8.I suggest showing, for example, a block diagram of the test facility with information on what quantities and where they were measured and at what frequency.
9. line 95-102 - Why is the temperature difference given in different units?
10. Where in formulas 1-3 was the power P used?
11. Was the usefulness of models other than a polynomial of degree three analyzed?
12. Why was a polynomial of degree three chosen? How was the significance of the parameters of the polynomial evaluated?  
13. table 2 - How was the observation divided into teaching and test data? Test set once 11 versus 22 observations?
14. fig. 7 and 8 - Different scales make interpretation difficult
15. fig. 9 - How was the cost of operating the chillers determined for each day?
16. I believe that two days is too short a period to verify the quality of the model. Please show that these are representative days for the whole year.
17. Can show the quality of the model for a learning set. How many days did the learning set cover?
18. No discussion and comparison of own research to other work in this area. What are the advantages and disadvantages of the developed method in relation to others.

I believe that the work needs to be improved before publication.

Author Response

(The authors gave the same response as above.)

Reviewer 3 Report

From my point of view, the work presents an interesting study. The results may have good acceptance among readers in areas for purposes. I consider that the document can be accepted in this version.

Author Response

(The authors gave the same response as above.)

Round 2

Reviewer 2 Report

The work after the correction is better. I still expect a change or comment to:
1. Specific heat of chilled water - wrong unit.
2. The review of the literature does not clearly indicate the need for further research in this area. Please indicate the disadvantages of the methods used in the process of controlling the operation of systems with air conditioning. Indicate what is new in this work.
3. How was the cost of operating the chillers determined for each day?
4. No discussion and comparison of own research to other work in this area. What are the advantages and disadvantages of the developed method in relation to others.

Author Response

Dear Sir:

Thank you for providing us the review’s comments. We have taken care of these precious comments and revised manuscript with the changes clearly identified by a highlighter pen. The point-to-point responses to all the referees are shown below.

  1. Specific heat of chilled water - wrong unit.

Ans.:The unit had been modified in pg.1.

  1. The review of the literature does not clearly indicate the need for further research in this area. Please indicate the disadvantages of the methods used in the process of controlling the operation of systems with air conditioning. Indicate what is new in this work.

Ans.:This manuscript had been described in pg.3.

  1. How was the cost of operating the chillers determined for each day?

Ans.: This manuscript used one day as a simulation example. The cost of operating the chillers determined is calculated based on power consumption of chillers. (in pg.4.)

  1. No discussion and comparison of own research to other work in this area. What are the advantages and disadvantages of the developed method in relation to others.

Ans.:The description of this problem is shown in pg.14.

I sincerely hope that we have clarified all your questions. Your assistance is very much appreciated. If you have further questions, please feel free to contact me.

Your assistance is highly appreciated.

Sincerely yours.

Dr. Ming-Tang Tsai

Department of E.E.,

Cheng-Shiu University

Email:k0217@gcloud.csu.edu.tw
